# ON QUANTIZING THE STATE OF THE MUON OPTIMIZER

## ABSTRACT

The Muon optimizer, based on matrix orthogonalization, has recently shown faster convergence and up to 2× computational efficiency over AdamW in LLM pretraining. Like AdamW, Muon is stateful, requiring storage of both model weights and accumulated gradients. While 8-bit AdamW variants mitigate this overhead using blockwise quantization, they are typically stable only under dynamic quantization - which improves stability on linear quantization for extreme values. In this paper, we introduce the 8-bit Muon optimizer using blockwise quantization, supporting both linear and dynamic schemes. We demonstrate that 8-bit Muon maintains stability under both, while delivering ∼74% reduction in memory footprint compared to full-precision Muon. In extensive experiments, 8-bit Muon with linear quantization outperforms AdamW and 8-bit AdamW in pre-training a 1.6B model on 4B FineWeb tokens, while achieving parity with Muon for Chinchilla-optimal training with 32B tokens for both validation loss and downstream benchmark tasks. It also shows competitive results when fine-tuning the Llama 3.2 3B model on post-training data. We also provide a theoretical perspective to help explain this robustness under quantization.

## 1 INTRODUCTION

Scaling laws for large language models (LLMs) (Kaplan et al., 2020; Hoffmann et al., 2022) indicate that larger models generally achieve better out-of-distribution performance across diverse tasks. Yet, GPU high-bandwidth memory (HBM) capacity has not kept pace with parameter counts. During training, memory is dominated by model parameters, gradients, optimizer states, and activations. Systems work has therefore focused on distributing these tensors across devices via distributed data parallel (DDP), Fully Sharded Data Parallel (FSDP) (Zhao et al., 2023), ZeRO stage-3 in Deep-Speed (Rajbhandari et al., 2020), and tensor/model parallelism (Shoeybi et al., 2019) in order to improve inference performance.

Orthogonal to sharding is compressing the optimizer state. AdamW (Loshchilov & Hutter, 2017; Kingma, 2014), the *de facto* optimizer for LLMs, maintains two FP32 moment buffers (first and second moments) per parameter. For an 8B-parameter model (e.g., an 8B Llama-3 variant (Dubey et al., 2024)), this alone occupies 64 GB (∼80% of an NVIDIA H100's 80 GB HBM), leaving little headroom for parameters, gradients, and activations. To mitigate this, Dettmers et al. (2021) quantize Adam's optimizer states to 8 bits via block-wise *dynamic* (non-linear) quantization, preserving stability in the presence of extreme values while reducing optimizer memory by roughly 4× — enabling performant training under tight memory budgets.

Recently, there has been a surge of interest in moving beyond AdamW to improve training efficiency (Anil et al., 2020; Shazeer & Stern, 2018; Vyas et al., 2024). Among various advances, one particularly promising optimizer is **Muon** (Jordan et al., 2024)[1], which orthogonalizes the gradient momentum before updating the parameters. Equalizing the importance of all update directions results in improved stability and better convergence (Bernstein & Newhouse, 2024a; Bernstein, 2025). Several large-scale studies have confirmed Muon's ability to achieve a 2x efficiency in a target validation loss compared to AdamW on a compute-optimal setup (Liu et al., 2025; Shah et al., 2025). Muon has also been used to train extremely large models up to a trillion parameters in size, like Kimi K2 (Team et al., 2025) and GLM4.5 (Zeng et al., 2025).

---

[1]Muon is closely related to SGD with momentum, adding a per-layer matrix orthogonalization

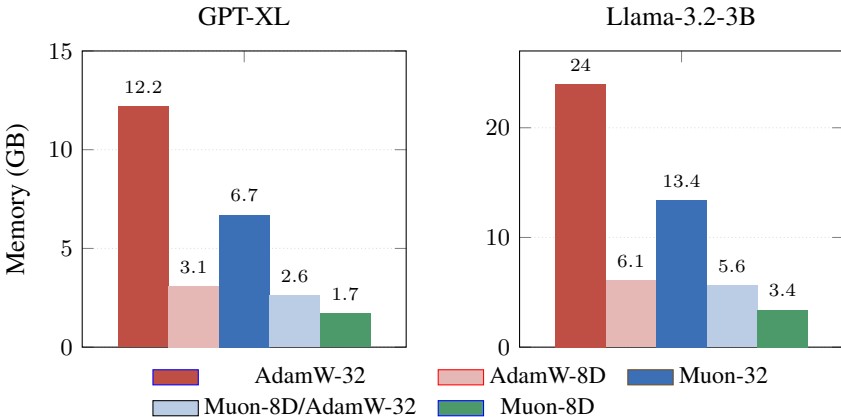

Figure 1: Optimizer state memory (GB) for GPT-XL (1.6B) and Llama-3.2-3B. `Muon-8D` reduces optimizer state for the GPT-XL and Llama models by as much as **86%** when compared to `AdamW-32`, and **74%** when compared to `Muon-32`. Table 1 defines the different variants in the legend.

In this paper, we introduce the **8-bit Muon** optimizer with blockwise quantization. While 8-bit AdamW variants are usually stable only using dynamic quantization, we demonstrate that **8-bit Muon can handle both linear and dynamic quantization effectively**. Our contributions can be summarized as:

- We study 8-bit AdamW, and pinpoint the source of its instability with linear quantization. Surprisingly, we also discover that SGD with momentum[2] and its closely related algorithm Muon are robust to linear quantization, achieving close to generalization parity when compared to their full-precision counterparts. **We also provide a theoretical perspective to explain SGD's and Muon's robustness under quantization.**.

- We **propose and evaluate several 8-bit variants of the Muon optimizer**, systematically applying both linear and dynamic quantization of Muon to hidden matrix-valued parameters and AdamW to the remaining parameters. Table 1 describes all the variants.

- **For pre-training** models up to 1.6B parameters in a Chinchilla-optimal fashion, we demonstrate that even simple 8-bit linear quantization for Muon is highly effective, **achieving virtually the same validation loss full-precision Muon optimizer for all model sizes**. For benchmarks, 8-bit linear quantization matches the quality of full-precision Muon for the 1.6B model, with a minor degradation at lower sizes.

- **For fine-tuning**, we train the Llama 3.2 3B model (Dubey et al., 2024) on splits of the `tulu-3-sft-mixture` post-training dataset (Lambert et al., 2024). **8-bit Muon demonstrates competitive performance when compared to Muon, AdamW and 8-bit AdamW variants**.

- Our 8-bit Muon variants reduce the optimizer state memory substantially. For the 1.6B model, an 8-bit variant has a state that is smaller by up to ∼**86% compared to AdamW (`Muon-8D` vs. `AdamW-32`)**, ∼**74% compared to Muon (`Muon-8D` vs. `Muon-32`)** and ∼**44% compared to 8-bit AdamW (`Muon-8D` vs. `AdamW-8D`)**; enabling efficient training of large models. We achieve a similar reduction in optimizer state for SFTing the Llama 3.2 3B model, **reducing the optimizer state footprint from ∼24 GB (`AdamW-32`) to ∼3.4 GB (`Muon-8D`)** (Figure 1).

---

[2]Throughout, we use SGD to mean SGD with momentum, unless otherwise noted.

| Model | Muon state | AdamW state | Shorthand |
|---|---|---|---|
| 32-bit AdamW | — | 32b | `AdamW-32` |
| 32-bit Muon | 32b | 32b | `Muon-32` |
| 8-bit AdamW (dynamic) | — | 8b D | `AdamW-8D` |
| 8-bit Muon (dynamic), 8-bit AdamW | 8b D | 8b D | `Muon-8D` |
| 8-bit Muon (linear), 8-bit AdamW | 8b L | 8b L | `Muon-8L` |
| 8-bit Muon (dynamic), 32-bit AdamW | 8b D | 32b | `Muon-8D/AdamW-32` |
| 8-bit Muon (linear), 32-bit AdamW | 8b L | 32b | `Muon-8L/AdamW-32` |

Table 1: Optimizer variants considered for pre-training and SFT. D = dynamic quantization, L = linear quantization. 8-bit Muon has 4 variants.

## 2 RELATED WORK

**Efficient Optimizers** There has been some prior work on alleviating the memory cost of training with optimizers like AdamW. Techniques like low-rank adaptation (LoRA) (Hu et al., 2022) allow a small subset of parameters to be fine-tuned for downstream tasks, but often fall behind in quality when compared to full parameter fine-tuning (Biderman et al., 2024). Adafactor (Shazeer & Stern, 2018) factorizes the second moment for matrices, reducing memory consumption compared to AdamW. Dettmers et al. (2021) introduced the 8-bit Adam optimizer, but it only works when combined with careful blockwise + dynamic quantization. Adam Galore (Zhao et al., 2024) leverages the low-rank structure of the gradients to reduce state size and can be combined with state quantization. Our work is directly comparable to Dettmers et al. (2021)'s work, and can potentially be combined with low-rank updates.

**Gradient Orthogonalization** The Muon optimizer (Jordan et al., 2024) has sparked interest in algorithms that take advantage of the orthonormalization of the gradient of matrix-valued parameters. Muon makes each weight matrix update orthonormal through polar decomposition (Newton-Schulz), giving direction-only, spectrally controlled steps for hidden layers. To scale it to large LLMs, recent works add decoupled weight decay and careful per-parameter update scaling (Liu et al., 2025). Dion (Ahn et al., 2025) leverages orthonormalization but is built for distributed training: using low-rank orthonormalization with device-local momentum/error-feedback to avoid reconstruction or synchronization of full matrices.

**Quantization** Quantization is a versatile tool for managing the memory cost of training large models. While we apply quantization to the state of the Muon optimizer, it has also been successfully applied to model weights in two settings - post-training quantization (PTQ) and quantization-aware training (QAT). PTQ uses calibration data to quantize the weights of large models in one shot to $k$ bits, where $k$ can be as low as 1 or 2 (Tseng et al., 2024; Frantar et al., 2022; Lin et al., 2024; Behdin et al., 2023). QAT involves training with quantized weights (Liu et al., 2023). Both PTQ and QAT require hardware support to realize the full benefits of quantization. Another interesting work is MuLoCo (Thérien et al., 2025), where the authors apply Muon as the inner (local) optimizer in a DiLoCo-style (Douillard et al.) loop, geared toward compressing parameter updates during distributed training. Thus, MuLoCo/DiLoco use quantization only for gradient updates.

## 3 BACKGROUND

### 3.1 QUANTIZATION FOR OPTIMIZERS

Quantization is the process of reducing the precision of numerical representations by mapping a value expressed in a richer way into a simpler one. For example, representing a real number as a 32-bit floating-point value, or converting a 32-bit float into an 8-bit integer, are both forms of quantization. In deep learning, model parameters and optimizer states are typically stored as 32-bit floating-point numbers, making their conversion to lower-precision formats a primary goal of quantization.

For instance, linear quantization is a method (among many) to quantize the state tensor $\mathbf{X}$ of an optimizer from 32-bit floats to 8-bit integers. This involves (a) dividing all the floats in the tensor by the absolute max to get a normalization constant $S = \max(|\mathbf{X}|)$ and (b) mapping the normalized

value to the integer $i$ in the range $-127$ to $127$, using uniform spacing of the normalized values in $[-1, 1]$, written as $i = \text{round}\left(\frac{\mathbf{X}}{S} \times 127\right)$. Dequantization then converts the codebook index back by multiplying with the previously stored normalization constant to recover $\widetilde{\mathbf{X}} = \frac{S}{127} \times i$.

For 8-bit quantization, the number of addressable states is 256. Dettmers et al. (2021) discuss another quantization codebook design - dynamic quantization. Dynamic quantization is a more sophisticated approach that quantizes non-uniformly by allocating more codes to regions with high densities and fewer codes to sparsely used regions. This increases resilience to non-uniform data.

A crucial recipe for reducing the effect of outliers is blockwise quantization (Dettmers et al., 2021), where one can segment a tensor into one or more blocks, and then perform quantization (linear, dynamic, or other schemes) separately within each block. A nice side effect of blockwise quantization is that each block can be processed in parallel. For a more detailed treatment of blockwise and dynamic quantization, please refer to (Dettmers et al., 2021).

## 3.2 THE MUON ALGORITHM

The Muon update on a single hidden layer can be described as follows:

$$
\begin{aligned}
\mathbf{M}^{(t)} &:= \beta\,\mathbf{M}^{(t-1)} + \nabla f_t\big(\mathbf{W}^{(t-1)}\big), \\
\mathbf{U}^{(t)} &:= \text{NS}\big(\mathbf{M}^{(t)}\big), \\
\mathbf{W}^{(t)} &:= \mathbf{W}^{(t-1)} - \alpha\,\mathbf{U}^{(t)}.
\end{aligned}
\tag{1}
$$

where $t$ is the iteration number and $\mathbf{M}$ is the momentum of the gradient. NS stands for the Newton-Schulz iteration process (Bernstein & Newhouse, 2024b; Higham, 2008), used to find an approximation for $\mathbf{U}\mathbf{V}^T$ where $\mathbf{U}\mathbf{\Sigma}\mathbf{V}^T$ is the singular value decomposition (SVD) of $\mathbf{M}$. Orthogonalization equalizes the importance of each update direction by collapsing all singular values to 1.

The vanilla version of Muon described above does not use weight decay. Additionally, it is not obvious whether it requires any hyperparameter tuning over AdamW baselines. Liu et al. (2025) introduced a variant of Muon that uses weight decay and also scales its update to match the update RMS of AdamW, producing the following version:

$$
\begin{aligned}
\mathbf{M}^{(t)} &:= \beta\,\mathbf{M}^{(t-1)} + \nabla f_t\big(\mathbf{W}^{(t-1)}\big), \\
\mathbf{U}^{(t)} &:= \text{NS}\big(\mathbf{M}^{(t)}\big), \\
\mathbf{W}^{(t)} &:= \mathbf{W}^{(t-1)} - \alpha\left(0.2 \cdot \mathbf{U}^{(t)} \cdot \sqrt{\max(m, n)} + \lambda\,\mathbf{W}^{(t-1)}\right).
\end{aligned}
\tag{2}
$$

where $m$ and $n$ are the dimensions of $\mathbf{M}$. Liu et al. (2025) claim that with Eq 2, hyperparameters such as learning rate and weight decay can be shared across matrix and non-matrix parameters. In the rest of the paper, any mention of Muon refers to the version in Equation 2, unless stated otherwise. It is important to note that any non-matrix parameters and input/output parameters are optimized using AdamW, leaving Muon to focus on matrix-valued hidden parameters. Complete algorithms can be found in Appendix B.

## 4 METHODS

### 4.1 WHY IS 8-BIT ADAMW UNSTABLE UNDER LINEAR QUANTIZATION?

Dettmers et al. (2021) observed that naïve 8-bit *linear* quantization performs poorly because it allocates too little resolution to small-magnitude entries, yielding large relative errors precisely where optimizer states concentrate most of their mass. We make this behavior concrete in Theorem 1 by analyzing how quantization error propagates and is magnified through the second-moment accumulator $\mathbf{v}$. In particular, we show that if moderate-size gradient coordinates occur with non-negligible probability, then the expected squared error of one step of Adam with linearly quantized states *diverges* as the numerical stabilizer $\epsilon \to 0$. The result holds for standard Adam hyperparameters; constants are kept explicit to emphasize practical regimes (e.g., $\epsilon \approx 10^{-8}$) where the error is already proved to be orders of magnitude larger than the unquantized update norm.

| Method | SGD+M | AdamW |
|--------|-------|-------|
| FP32 | 76.21 | 74.42 |
| 8-bit linear quant. | 76.25 | — |

Table 2: Top-1 validation accuracy (%) after 90 epochs for SGD+M and AdamW in FP32 and with 8-bit linear quantization. "—" indicates that Adam with linear quantization diverged.

Let $Q$ denote the 8-bit linear quantization operator from Definition 1. We analyze the base Adam algorithm without weight decay[3]. The quantized variant applies $Q$ to the moment estimates before forming the update. All algorithmic details, definitions, and proofs are provided in Appendix C.

**Theorem 1** *Let $\boldsymbol{\theta}^{(1)}$ denote the parameters after one step of Adam as given in Algorithm 4, and let $\tilde{\boldsymbol{\theta}}^{(1)}$ denote the parameters after one step of the same algorithm with 8-bit linear quantization applied to the moment estimates (Definition 1), i.e.:*

$$\tilde{\boldsymbol{\theta}}^{(1)} = \boldsymbol{\theta}^{(0)} - \alpha \cdot \frac{Q(\mathbf{m}^{(1)})}{\sqrt{Q(\mathbf{v}^{(1)})} + \epsilon}.$$

*Suppose that each entry of $\mathbf{g}^{(1)} \in \mathbb{R}^d$ satisfies $\mathbb{P}\left( \frac{\|\mathbf{g}\|_\infty}{60} < |\mathbf{g}_i| < \frac{\|\mathbf{g}\|_\infty}{16} \right) \geq \nu$ and $\|\mathbf{g}\|_\infty \geq g_\infty > 256\,\epsilon$ with probability one. Then*

$$\mathbb{E}\,\|\boldsymbol{\theta}^{(1)} - \tilde{\boldsymbol{\theta}}^{(1)}\|_2^2 \;\geq\; \frac{d\nu\,\alpha^2 g_\infty^2}{(256\,\epsilon)^2}.$$

### 4.2 THE CURIOUS CASE OF 8-BIT SGD WITH MOMENTUM

The proof of Theorem 1 shows that the instability of Adam with linear quantization arises primarily from error in the second-moment vector, which appears in the denominator of the update rule. This naturally raises the question: **does 8-bit linear quantization suffice when such a denominator is avoided?**

From a theoretical perspective, we show that, unlike Adam, SGD with momentum admits a uniform error bound under linear quantization. In particular, for any initialization of the weights and momentum, the quantization error remains bounded. Let $\eta > 0$ denote the step size and $\rho \in [0, 1)$ the momentum parameter.

**Theorem 2** *Consider a step of SGD with momentum with and without 8-bit linear quantization of the momentum:*

$$\tilde{\boldsymbol{\theta}}^{(t+1)} = \boldsymbol{\theta}^{(t)} - \eta(\mathbf{g}^{(t)} + \rho Q(\mathbf{m}^{(t)})) \;\; and \;\; \boldsymbol{\theta}^{(t+1)} = \boldsymbol{\theta}^{(t)} - \eta(\mathbf{g}^{(t)} + \rho\mathbf{m}^{(t)}).$$

*From any point $\boldsymbol{\theta}^{(t)}$ and any momentum state $\mathbf{m}^{(t)}$, if $Q$ is as in Definition 1, then*

$$\|\tilde{\boldsymbol{\theta}}^{(t+1)} - \boldsymbol{\theta}^{(t+1)}\|_2^2 \leq d\eta^2\rho^2 \left( \frac{\|\mathbf{m}^{(t)}\|_\infty}{127} \right)^2.$$

Empirically, we confirm this difference. We train a ResNet-50 model (He et al., 2016) on the ImageNet dataset (Deng et al., 2009), using a standard training regime of 90 epochs. We compare AdamW and variants of SGD with momentum. Results are in Table 2. **Surprisingly, SGD with linear 8-bit quantization achieves the same high validation top-1 accuracy (Goyal et al., 2017) of 76%+ as full-precision SGD**. AdamW underperforms when compared to SGD (a well-known result on image classification training), while AdamW with linear quantization diverges immediately. See Appendix A.1.1 for extended details.

Together, these theoretical and empirical results demonstrate that the instability of quantized Adam is specifically driven by the quantization of the second-moment term in the denominator.

---

[3]The result extends immediately to AdamW, since the decay of $\boldsymbol{\theta}^{(0)}$ affects both $\tilde{\boldsymbol{\theta}}^{(1)}$ and $\boldsymbol{\theta}^{(1)}$ equally and therefore cancels out.

### 4.3 THE 8-BIT MUON ALGORITHM

Since SGD works well with linear quantization, we now propose 8-bit quantized variants for the Muon algorithm that leverage either linear or dynamic schemes. Because Muon, like SGD, doesn't use a second-order momentum term, we hypothesize it will train stably with linear quantization. In Table 1, we introduce various variants of Muon and AdamW that leverage different quantization schemes. Since Muon uses orthogonalization only for hidden matrix-valued parameters, the other parameters like embeddings and classifier heads are usually optimized with AdamW. The variants listed in Table 1 consist of linear, dynamic, and even hybrid versions.

The 8-bit Muon update can be written as:

$$
\begin{aligned}
\widetilde{\mathbf{M}}^{(t-1)} &:= \mathrm{DQ}_B^{\mathrm{mode}}\big(\mathbf{Z}^{(t-1)}, \mathcal{S}^{(t-1)}\big), \\
\mathbf{M}^{(t)} &:= \beta\,\widetilde{\mathbf{M}}^{(t-1)} + \nabla f_t\big(\mathbf{W}^{(t-1)}\big), \\
\mathbf{U}^{(t)} &:= \mathrm{NS}\big(\mathbf{M}^{(t)}\big), \\
\mathbf{W}^{(t)} &:= (1 - \alpha\lambda)\,\mathbf{W}^{(t-1)} \; - \; 0.2\,\alpha\,\sqrt{\max(m,n)}\,\mathbf{U}^{(t)}, \\
\big(\mathbf{Z}^{(t)}, \mathcal{S}^{(t)}\big) &:= \mathrm{Q}_B^{\mathrm{mode}}\big(\mathbf{M}^{(t)}\big).
\end{aligned}
\tag{3}
$$

where $\mathbf{Z}$ refers to the compressed momentum buffer and $\mathcal{S}$ is the associated state required to dequantize it. $\mathrm{Q}_B^{\mathrm{mode}}$ is the quantization function, and $\mathrm{DQ}_B^{\mathrm{mode}}$ dequantizes the compressed momentum vector for use in the update. Rest of the notation is borrowed from Equation 2. The complete algorithms for 8-bit Muon can be found in Appendix B.

We now claim that like SGD, even Muon admits a uniform bound under linear quantization. In fact, in Theorem 3 we show the following: when Muon uses an exact orthogonalization procedure (via the SVD), the quantization error bound for a single layer matches the SGD case up to an additional dependence on the smallest singular value, $s$, of the momentum matrix. This dependence is natural, since the conditioning of the momentum controls the stability of the orthogonal factor. In practice, the margin $s$ is typically not small, so the guarantee has the same qualitative form as the SGD result.

**Theorem 3** *Consider a step of Muon with momentum (using the exact polar factor rather than the Newton–Schulz approximation) with and without 8-bit linear quantization of the momentum. Let the layer weights and momentum state be $\mathbf{W}^{(t-1)}$ and $\mathbf{M}^{(t-1)}$, each with $d$ entries (see Appendix C.4 for full update formulas).*

*Suppose that, after a single gradient update, both the original and quantized momentum matrices are full column rank with minimum singular value at least $s > 0$. Then*

$$
\|\widetilde{\mathbf{W}}^{(t)} - \mathbf{W}^{(t)}\|_{\mathrm{F}}^2 \; \leq \; \frac{d\alpha^2\beta^2}{s^2}\left(\frac{\|\operatorname{vec}(\mathbf{M}^{(t-1)})\|_\infty}{127}\right)^2,
$$

*where $\widetilde{\mathbf{W}}^{(t)}$ denotes the weights after the quantized update and $\mathbf{W}^{(t)}$ after the unquantized update.*

We note that the empirical validation of Theorems 1, 2 and 3 can be found in Appendix C.5.

## 5 EXPERIMENTS

### 5.1 PRE-TRAINING WITH 8-BIT MUON

**Architectures** For the pre-training task, we train a modified version of the GPT2 architecture (Radford et al., 2019) from scratch, in which the learned positional embeddings are replaced by rotary positional embeddings (RoPE) (Su et al., 2024). To understand the scaling effect, we consider 3 different sizes - Medium (405M), Large (834M) and XL (1.6B). For the sake of brevity, we will refer to this architecture as GPT in the rest of the paper. Detailed notes on the architecture can be found in Table 7 in Appendix A.1.2.

**Datasets** Our pre-training dataset consists of the FineWeb-Edu dataset Penedo et al. (2024). Based on the findings of Hoffmann et al. (2022), we use approximately 20 tokens per parameter, leading

to a training set that scales with the model size (often referred to as Chinchilla optimal in literature). Our largest training dataset consists of approximately 32 billion FineWeb tokens for the 1.6B model. We use 150k samples from the validation split of FineWeb to measure validation loss, totaling approximately 300M tokens.

**Training details** We pre-train the 3 GPT models from scratch using data from FineWeb described earlier. For AdamW, we set $\beta_1$ and $\beta_2$ to 0.9 and 0.999 respectively. The $\epsilon$ value for AdamW variants is set to $10^{-8}$. For Muon, we set the momentum parameter to 0.95.

The block size of the quantized versions of AdamW and Muon is set to 2048. We use the bitsand-bytes library (Dettmers et al., 2023) for quantization / dequantization routines for linear and dynamic quantization. We fix decoupled weight decay to 0.1 for all experiments.

We use the WSD learning rate schedule (Hu et al., 2024), with a linear warmup of the learning rate from 0 to the peak in 10% of training steps, a linear decay to zero in the last 10% steps. For each model size, we use the peak learning rate and global batch size as prescribed by Brown et al. (2020), since they tuned it for various sizes of the GPT architecture for the AdamW optimizer. For Muon, we make no attempt to tune the peak learning rate for its variants and re-use the peak learning rates used for AdamW, since **our version of Muon is supposed to be a drop-in replacement for AdamW without any requirement to tune learning rate or weight decay** (Liu et al., 2025).

We use distributed data parallel (DDP) for all experiments, with training working on multiple GPUs at the same time. Our largest model is 1.6B and fits on a single NVIDIA B200 GPU. The global batch size per model size is achieved with varying gradient accumulation.

A detailed table on hyperparameters and training configurations can be found in Appendix A.1.2.

**Evaluation criteria** Our evaluation of pre-training is two-pronged - (a) validation loss and (b) benchmark performance on six different tasks using the lm-eval harness (Gao et al., 2024). These tasks include MMLU (Hendrycks et al., 2020), LAMBADA (Paperno et al., 2016), BoolQ (Clark et al., 2019), HellaSwag (Zellers et al., 2019), ARC-Challenge and ARC-Easy (Clark et al., 2018).

### 5.1.1 RESULTS

**Investigating the stability of 8-bit Muon optimizers** In Figure 2, we show the results of comparing all variants of 8-bit Muon described in Table 1 on the task of pre-training the GPT-Medium model. **Before testing these variants, we confirmed that `AdamW-8L` diverged on this problem early during training**. Interestingly, `Muon-8L` did not diverge, but performed poorly. Remarkably, all other versions of 8-bit Muon closely match each other. This includes `Muon-8L/AdamW-32`, a version that uses linear quantization for Muon-associated parameters. **This corroborates our theoretical findings that AdamW is particularly unstable especially when using simple techniques like linear quantization. Because of these findings, we avoid the `Muon-8L` variant and instead use alternatives like `Muon-8L/AdamW-32`.**

Our main results consists of two main parts, described in Figure 3 and Tables 3 and 4. We make the following observations:

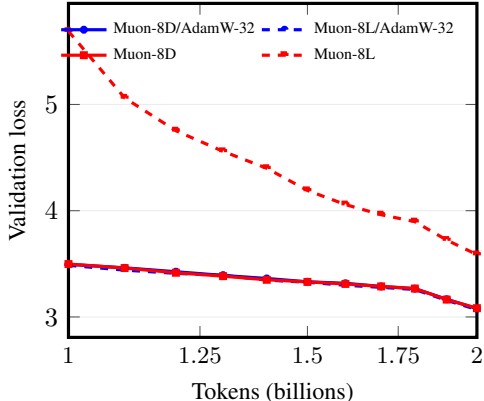

Figure 2: Training curves on GPT-Medium comparing four variants of 8-bit Muon (zoomed-in version, from 1B to 2B tokens). Except `Muon-8L`, all the others follow each other closely.

**Muon outperforms AdamW variants and 8-bit Muon variants are competitive with Muon**

In the first set of pre-training experiments, we aim to comparing various versions of AdamW and Muon on the same task. To this end, we compare their performance on 4 billion tokens of the

FineWeb dataset for the XL architecture for two reasons - (a) to understand which variants of Muon, if any, outperform AdamW and (b) to compare various versions of Muon to each other.

The results of the experiment be found in the top left subfigure in Figure 3 and the first row of Table 3. Unsurprisingly, all variants of Muon outperform all variants of AdamW. This is despite no Muon-specific tuning of the peak learning rate or weight decay.

Surprisingly, all the 8-bit variants of Muon achieve a loss comparable to that of full precision Muon, comfortably outperforming `AdamW-32` and `Adam-8D`. This includes `Muon-8L/AdamW-32`, which uses linear quantization for hidden matrix-valued parameters. **These results show that Muon is robust to the underlying quantization scheme for state compression**.

**`Muon-32` and `Muon-8L/AdamW-32` are comparable for various model sizes**

After having established that Muon variants outperform AdamW and 8-bit Muon variants are quite similar to each other, the second set of our pretraining experiments focus on comparing `Muon` and `Muon-8L/AdamW-32`. **We make this choice because linear quantization is inherently more challenging to scale to billion-plus parameter models**.

We compare these two optimizers for Chinchilla-optimal training of the medium, large and XL GPT models. Figure 3 and Table 3 reveal that both optimizers are virtually indistinguishable in terms of the final validation loss achieved. From Table 4, we note that the quantized model has a slight degradation for the medium and large sizes on downstream tasks, and is at parity for the XL model. **This proves that quantized versions of Muon can perform competitively with full-precision Muon for model pre-training, achieving parity for the XL model for Chinchilla-optimal training**.

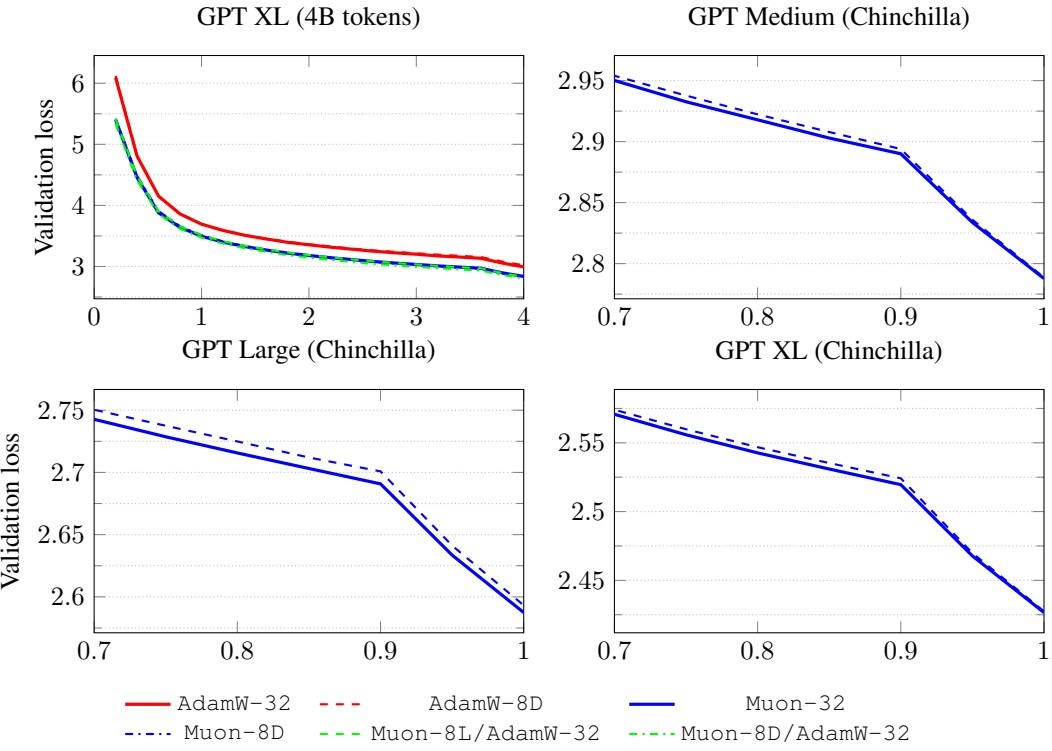

Figure 3: Validation loss comparison. **Top-left**: GPT XL with six optimizers over 4B tokens. **Other plots**: Chinchilla-optimal training comparing `Muon-32` vs `Muon-8L/AdamW-32`, zoomed to last 30% of training. 8-bit Muon closely matches full-precision Muon across all scales.

| Model | A-32 | A-8D | Muon-32 | M-8L/A-32 loss | (% close) | M-8D/A-32 loss | (% close) | M-8D loss | (% close) |
|---|---|---|---|---|---|---|---|---|---|
| XL (4B) | 2.994 | 3.020 | 2.837 | 2.830 | $-0.25$ | 2.822 | $-0.53$ | **2.803** | $-1.20$ |
| *Chinchilla-optimal (Muon-32 vs Muon-8L/A-32 only):* | | | | | | | | | |
| Medium (C) | – | – | **2.788** | 2.789 | 0.04 | – | – | – | – |
| Large (C) | – | – | **2.587** | 2.593 | 0.23 | – | – | – | – |
| XL (C) | – | – | **2.427** | **2.427** | 0.00 | – | – | – | – |

Table 3: Final validation loss across models and optimizers. M = Muon, A = AdamW. **Top**: XL trained on 4B tokens with all six optimizers. All 8-bit Muon variants closely match full-precision Muon. **Bottom**: Chinchilla-optimal training (C) comparing only `Muon-32` vs `Muon-32/AdamW-32`.

| | Medium | | Large | | XL | |
|---|---|---|---|---|---|---|
| Task | M-32 | M-8L/A-32 | M-32 | M-8L/A-32 | M-32 | M-8L/A-32 |
| ARC-Challenge | 0.273 | 0.269 | 0.294 | 0.281 | 0.324 | 0.329 |
| ARC-Easy | 0.473 | 0.465 | 0.527 | 0.514 | 0.603 | 0.591 |
| BoolQ | 0.616 | 0.573 | 0.602 | 0.591 | 0.586 | 0.600 |
| HellaSwag | 0.343 | 0.344 | 0.399 | 0.399 | 0.502 | 0.500 |
| LAMBADA | 0.236 | 0.227 | 0.312 | 0.299 | 0.387 | 0.384 |
| MMLU | 0.230 | 0.230 | 0.237 | 0.233 | 0.247 | 0.249 |
| **Avg. (6 tasks)** | **0.362** | **0.351** | **0.395** | **0.386** | **0.441** | **0.442** |

Table 4: Downstream task accuracy for Chinchilla-optimal models. M = Muon, A = AdamW. `Muon-8L/AdamW-32` closely matches full-precision Muon across all tasks for the XL model, and has a slight drop for the medium and large models.

## 5.2 FINE-TUNING WITH 8-BIT MUON

**Architectures** We use the Llama 3.2 3B base model (Dubey et al., 2024) for all our fine-tuning (SFT) experiments. It is empirically well-known that if a model is pre-trained with one optimizer, then empirically it is best to fine-tune it with the same optimizer Liu et al. (2025); Team et al. (2025). **Our aim is to assess how closely quantized versions of AdamW and Muon can match their full precision counterparts, and not to achieve state-of-the-art results.**

**Datasets** We use the `tulu-3-sft-mixture` dataset (Lambert et al., 2024) - an open dataset for post-training which targets a diverse set of skills such as reasoning and math. The dataset consists of close to 1 million training samples.

**Training Details** We use SFT on the Llama model using all the optimizers used for pre-training. For each optimizer, we use a random 10k split samples from the full Tulu-3 dataset to perform lightweight fine-tuning, since our aim is to compare optimizers in a fair setting. This data-constrained setup is ideal for our aim of comparing optimizers across varying training regimes. Since SFT of LLMs can have non-negligible variance, we conduct 5 runs per experiment.

We inherit all the optimizer setup from pre-training, except for learning rate and weight decay. Learning rate follows a linear warmup to $10^{-5}$ for the first 3% steps, followed by linear decay to 0. Weight decay is set to 0.01. We use a single NVIDIA H100 GPU for all experiments, with a batch size of 2 and gradient accumulation of 8, yielding a global batch size of 16 sequences.

**Evaluation Criteria** Our SFT evaluation is performed on the two benchmarks: HumanEval(pass@1) (Chen et al., 2021) for coding and GSM8K (Cobbe et al., 2021) for math.

### 5.2.1 RESULTS

SFT results can be found in Table 5. Again, the 8-bit variants of Muon match the performance of the 32-bit version. Since Llama 3.2 3B is pre-trained with AdamW, there is no clear winner between AdamW and Muon, mirroring findings from other papers (Liu et al., 2025; Team et al., 2025). We

| Model (GPT) | Adam-32 | Adam-8D | Muon-32 | M-8L/A-32 | M-8D/A-32 | M-8D |
|---|---|---|---|---|---|---|
| GSM8K | $28.300_{0.61}$ | $29.160_{0.42}$ | $28.050_{0.425}$ | $28.660_{0.619}$ | $28.253_{0.57}$ | $28.730_{0.35}$ |
| HumanEval | $26.630_{0.70}$ | $27.640_{1.86}$ | $26.420_{0.93}$ | $27.640_{0.35}$ | $26.830_{0.61}$ | $26.950_{1.17}$ |

Table 5: Llama 3.2 3B SFT Results after lightweight training. Standard deviation across 5 runs is shown in subscript. 8-bit variants are competitive with their 32-bit versions.

expect Muon-based fine-tuning to be increasingly beneficial as more models pre-trained with Muon are released.

### 5.3 MEMORY FOOTPRINT

Table 6 compares the persistent HBM memory footprint of the optimizer state for variants, when profiled for the GPT Small, Medium and XL models, as well as Llama-3.2-3B Grattafiori et al. (2024) (Figure 1 summarizes the same info for XL and Llama). For the biggest models like XL and Llama, `Muon-8D` provides substantial relative savings of $\sim$**74%**, $\sim$**86%** and $\sim$**44%**, when compared to `Muon-32`, `AdamW-32` and `AdamW-8D` respectively.

| Model | Adam-32 | Adam-8D | Muon-32 | M-8L/A-32 | M-8D |
|---|---|---|---|---|---|
| XS | 0.73 | 0.19 | 0.58 | 0.47 | **0.15** |
| Small | 1.22 | 0.31 | 0.90 | 0.66 | **0.23** |
| Medium | 3.02 | 0.77 | 1.89 | 1.05 | **0.47** |
| XL | 12.19 | 3.10 | 6.69 | 2.58 | **1.68** |
| Llama-3.2-3B | 23.94 | 6.10 | 13.44 | 5.58 | **3.37** |

Table 6: Optimizer states memory (GB) across all model sizes tested (lower is better). `Muon-8D` reduces optimizer memory footprint for GPT XL and Llama substantially.

For smaller models, `Muon-8L/AdamW-32` has a larger optimizer state memory footprint than `Adam-8D`, a trend that inverts as model size increases. This is because the size of the embedding and lm-head matrices remains the same across model sizes because of a fixed vocabulary. For the XS model, these layers constitute up to 47% of model parameters and are optimized with 32-bit AdamW. The high memory cost for these layers offsets the savings achieved in the rest of the model.

## 6 CONCLUSIONS

In this paper, we introduced 8-bit Muon, a memory-efficient optimizer designed to address the problem of large memory footprints of LLMs. We build on the Muon optimizer and leverage blockwise quantization. One key finding is the robustness of Muon to types of quantization. Our results across pre-training and fine-tuning of large models show that our 8-bit Muon variants nearly matched the performance of the full-precision Muon. In terms of practical benefits, our method reduced the optimizer state memory by up to 86% compared to AdamW and 74% compared to full-precision Muon for models up 1.6B-3B in size. Future work could include quantization to even lower bits, as well as combination with techniques like low-rank matrices.

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

## A  APPENDIX

### A.1  TRAINING HYPERPARAMETER DETAILS

#### A.1.1  IMAGENET TRAINING DETAILS

We trained ResNet-50 on ImageNet for 90 epochs using two H100 GPUs with PyTorch Distributed-DataParallel. The schedule was the standard 90-epoch multi-step regime with learning rate decays

| Model size | Parameters | $d_{model}$ | $n_{layers}$ | $n_{heads}$ | FF Ratio |
|---|---|---|---|---|---|
| Medium | 405M | 1024 | 24 | 16 | 4 |
| Large | 834M | 1536 | 24 | 16 | 4 |
| XL | 1.6B | 1600 | 48 | 25 | 4 |

Table 7: Model sizes used for the pre-training task.

at epochs 30, 60, and 80. For SGD with momentum we used a batch size of 128 per GPU (256 total), momentum 0.9, and weight decay $10^{-4}$. For AdamW we used a learning rate of $3 \times 10^{-3}$ and weight decay $10^{-2}$. Hyperparameters were held fixed across FP32 and quantized runs. Training images were augmented with random resized crops to $224 \times 224$ and random horizontal flips. At evaluation time, images were resized to 256 pixels on the short side and center-cropped to $224 \times 224$, followed by normalization with the standard ImageNet mean and variance.

In the quantized variants, optimizer states were stored in 8-bit linear form with per-tensor scaling (Definition 1 applied layer-wise). For SGD with momentum, only the momentum buffer was quantized. For AdamW, both the first- and second-moment estimates were quantized. At each step, stored values were dequantized for computation, updated, and then requantized. Model weights, gradients, and activations were always maintained in FP32.

Figure 4 reports validation accuracy during training. Quantized SGD matches the FP32 baseline throughout. AdamW with FP32 optimizer states achieves slightly lower accuracy, while the quantized AdamW variant diverged immediately and is not shown.

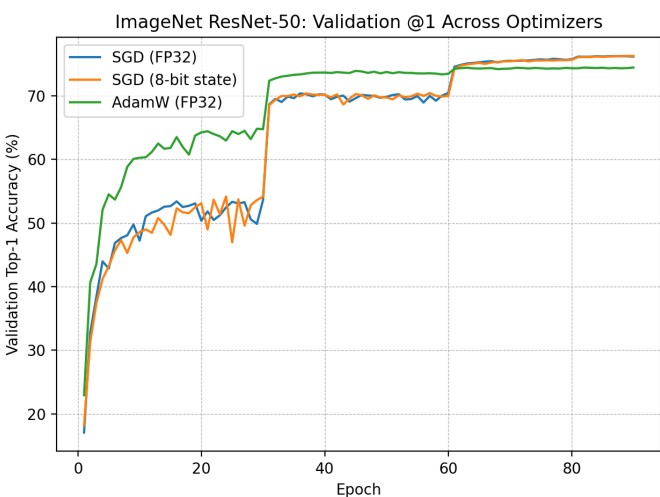

Figure 4: Validation top-1 accuracy on ImageNet for ResNet-50. Quantized SGD overlaps with the FP32 baseline. AdamW with FP32 states underperforms slightly, while the quantized AdamW variant diverged at the first step and is not shown.

### A.1.2   PRE-TRAINING DETAILS

We train 3 GPT-style models using 8 B200 GPUs with PyTorch Distributed DataParallel (DDP). A summary of the model architectures used is described in Table 7. A summary of the training hyperparameters used for training is shown in Table 8

## B   MUON ALGORITHMS

Algos 1, 2 and 3 represent vanilla Muon (Jordan et al., 2024), Muon (Liu et al., 2025) and Quantized Muon. **Note**: The algorithms use lowercase vector notation.

| Model size | LR | Local BS | Grad. Acc. Steps | Global BS |
|---|---|---|---|---|
| Medium | $3 \cdot 10^{-4}$ | 32768 | 1 | 0.5M |
| Large | $2.5 \cdot 10^{-4}$ | 32768 | 1 | 0.5M |
| XL | $2 \cdot 10^{-4}$ | 16384 | 4 | 1M |

Table 8: Training configuration for the pre-training task. LR refers to the learning rate, and BS refers to the batch size. Both local and global batch sizes reported are in number of tokens, as all models were trained with a context length of 2048.

---

**Algorithm 1** Vanilla Muon (using $\boldsymbol{\theta}, \mathbf{g}, \mathbf{m}$; NS-orthogonalized momentum for 2D hidden-layer params)

---

1: **Input:** step size $\alpha$, momentum $\beta \in [0, 1)$, NS steps $T$ (default 5), $\epsilon > 0$
2: **Initialize:** $\boldsymbol{\theta}^{(0)}, \tilde{\mathbf{m}}^{(0)} = \mathbf{0}$
3: **for** $t = 1, 2, \ldots$ **do**
4:     $\mathbf{g}^{(t)} \leftarrow \nabla f_t(\boldsymbol{\theta}^{(t-1)})$
5:     $\tilde{\mathbf{m}}^{(t)} \leftarrow \beta \tilde{\mathbf{m}}^{(t-1)} + (1 - \beta) \mathbf{g}^{(t)}$          ▷ SGD-style momentum *before* orthogonalization
6:     **for each** 2D hidden-layer parameter block $\theta \subset \boldsymbol{\theta}$ **do**
7:         $\Delta \leftarrow \text{NS}(\tilde{\mathbf{m}}^{(t)}[\theta], T, \epsilon)$          ▷ Orthogonalize update
8:         $\theta \leftarrow \theta - \alpha \Delta$
9:     **end for**
10: **end for**

---

---

**Algorithm 2** Muon (using $\boldsymbol{\theta}, \mathbf{g}, \mathbf{m}$; NS-orthogonalized momentum for 2D hidden-layer params)

---

1: **Input:** step size $\alpha$, momentum $\beta \in [0, 1)$, weight decay $\lambda \geq 0$, NS steps $T$ (default 5), $\epsilon > 0$
2: **Initialize:** $\boldsymbol{\theta}^{(0)}, \tilde{\mathbf{m}}^{(0)} = \mathbf{0}$
3: **for** $t = 1, 2, \ldots$ **do**
4:     $\mathbf{g}^{(t)} \leftarrow \nabla f_t(\boldsymbol{\theta}^{(t-1)})$
5:     $\tilde{\mathbf{m}}^{(t)} \leftarrow \beta \tilde{\mathbf{m}}^{(t-1)} + (1 - \beta) \mathbf{g}^{(t)}$          ▷ SGD-style momentum *before* orthogonalization
6:     **for each** 2D hidden-layer parameter block $\theta \subset \boldsymbol{\theta}$ (with dimensions $m, n$) **do**
7:         $\Delta \leftarrow \text{NS}(\tilde{\mathbf{m}}^{(t)}[\theta], T, \epsilon)$          ▷ Orthogonalize update
8:         $\theta \leftarrow (1 - \alpha\lambda)\theta - 0.2\alpha\sqrt{\max(m, n)}\,\Delta$
9:     **end for**
10: **end for**

---

---

**Algorithm 3** Quantized Muon (using $\boldsymbol{\theta}, \mathbf{g}, \mathbf{m}$; NS-orthogonalized momentum for 2D hidden-layer params)

---

1: **Input:** step size $\alpha$, momentum $\beta \in [0, 1)$, weight decay $\lambda \geq 0$, NS steps $T$, $\epsilon > 0$, quantization params (mode, bits $B$)
2: **Initialize:** $\boldsymbol{\theta}^{(0)}$, quantized momentum state $(\mathbf{z}^{(0)}, \mathbf{s}^{(0)}) = (\mathbf{0}, \mathbf{0})$
3: **for** $t = 1, 2, \ldots$ **do**
4:     $\mathbf{g}^{(t)} \leftarrow \nabla f_t(\boldsymbol{\theta}^{(t-1)})$
5:     **for each** 2D hidden-layer parameter block $\theta \subset \boldsymbol{\theta}$ (with dimensions $m, n$) **do**
6:         $\tilde{\mathbf{m}}^{(t-1)} \leftarrow \text{DEQUANTIZE}(\mathbf{z}^{(t-1)}[\theta], \mathbf{s}^{(t-1)}[\theta])$          ▷ Dequantize saved momentum state
7:         $\mathbf{m}^{(t)} \leftarrow \beta\,\tilde{\mathbf{m}}^{(t-1)} + \mathbf{g}^{(t)}[\theta]$          ▷ Update momentum with current gradient
8:         $\mathbf{u}^{(t)} \leftarrow \text{NS}(\mathbf{m}^{(t)}, T, \epsilon)$          ▷ Orthogonalize momentum for the update
9:         $\theta \leftarrow (1 - \alpha\lambda)\theta - 0.2\alpha\sqrt{\max(m, n)}\,\mathbf{u}^{(t)}$          ▷ Update parameters
10:         $(\mathbf{z}^{(t)}[\theta], \mathbf{s}^{(t)}[\theta]) \leftarrow \text{QUANTIZE}(\mathbf{m}^{(t)})$          ▷ Quantize and save new momentum state
11:     **end for**
12: **end for**

---

## C  QUANTIZATION ERROR BOUNDS

In this appendix we formalize the 8-bit linear quantization operator and provide detailed proofs of the error bounds for Adam and SGD with momentum under quantization. We adopt the same notation as in Algorithm 4: $\mathbf{g}$ is the stochastic gradient, $\mathbf{m}$ and $\mathbf{v}$ are the first- and second-moment accumulators, $\mathbf{g}^2$ denotes the entrywise square, $\sqrt{\mathbf{v}}$ denotes the entrywise square root, and $\text{vec}(\mathbf{M})$ denotes the vectorization of $\mathbf{M}$.

### C.1  QUANTIZATION OPERATOR

**Definition 1** *(Linear quantization). For a given vector $\mathbf{x} \in \mathbb{R}^d$ denoting the optimizer state of some algorithm, the 8-bit linear quantization is denoted by $Q : \mathbb{R}^d \to \mathbb{R}^d$, where:*

$$[Q(\mathbf{x})]_i = \frac{\|\mathbf{x}\|_\infty}{127} \cdot \text{round}\left(\frac{127 \cdot \mathbf{x}_i}{\|\mathbf{x}\|_\infty}\right).$$

That is, each coordinate of $\mathbf{x}$ is mapped to the nearest grid point in a uniform partition of $[-\|\mathbf{x}\|_\infty, \|\mathbf{x}\|_\infty]$ into 256 representable levels (corresponding to signed 8-bit integers from $-128$ to 127), then rescaled back to floating point. This is the standard max-abs scaling scheme used in prior 8-bit quantization work. Note that, although we work over the reals, this definition effectively models the quantization and de-quantization steps applied to optimizer states between iterations.

### C.2  PROOFS

#### C.2.1  PROOF OF THEOREM 1

**Proof**

As stated in the theorem, we work at the first step ($t = 1$) and suppress iterate superscripts for notational clarity. By the moment definitions in Algorithm 4, we have $\mathbf{m} = \mathbf{g}$ and $\mathbf{v} = \mathbf{g}^2$ (entrywise), so for each coordinate $i$ we have $\sqrt{\mathbf{v}_i} = |\mathbf{g}_i|$. We first lower bound the per-coordinate deviation $\left|\frac{\mathbf{m}_i}{\sqrt{\mathbf{v}_i}+\epsilon} - \frac{Q(\mathbf{m}_i)}{\sqrt{Q(\mathbf{v}_i)}+\epsilon}\right|$ and then sum over $i$.

$$\mathbb{P}\left(\left(\frac{\mathbf{m}_i}{\sqrt{\mathbf{v}_i}+\epsilon} - \frac{Q(\mathbf{m}_i)}{\sqrt{Q(\mathbf{v}_i)}+\epsilon}\right)^2 \geq t^2\right) = \mathbb{P}\left(\left|\frac{\mathbf{m}_i}{\sqrt{\mathbf{v}_i}+\epsilon} - \frac{Q(\mathbf{m}_i)}{\sqrt{Q(\mathbf{v}_i)}+\epsilon}\right| \geq t\right)$$

$$\geq \mathbb{P}\left(\frac{|\mathbf{g}_i| - \|\mathbf{g}\|_\infty/127}{\sqrt{Q(\mathbf{v}_i)}+\epsilon} - \frac{|\mathbf{g}_i|}{\sqrt{\mathbf{v}_i}+\epsilon} \geq t \text{ and } \mathbf{v}_i \geq Q(\mathbf{v}_i)\right)$$

Note that $\sqrt{\mathbf{v}_i} = \sqrt{\mathbf{g}_i^2} = |\mathbf{g}_i|$ and $Q(\mathbf{v}_i) = 0$ implies $\mathbf{v}_i \geq Q(\mathbf{v}_i)$. Hence, following from above,

$$\mathbb{P}\left(\left(\frac{\mathbf{m}_i}{\sqrt{\mathbf{v}_i}+\epsilon} - \frac{Q(\mathbf{m}_i)}{\sqrt{Q(\mathbf{v}_i)}+\epsilon}\right)^2 \geq t^2\right) \geq \mathbb{P}\left(\frac{|\mathbf{g}_i| - \|\mathbf{g}\|_\infty/127}{\epsilon} - \frac{|\mathbf{g}_i|}{|\mathbf{g}_i|+\epsilon} \geq t \text{ and } Q(\mathbf{v}_i) = 0\right)$$

$$\geq \mathbb{P}\left(\frac{|\mathbf{g}_i| - \|\mathbf{g}\|_\infty/127}{\epsilon} - 1 \geq t \text{ and } Q(\mathbf{v}_i) = 0\right)$$

Define the event $E_i := \{\frac{\|\mathbf{g}\|_\infty}{60} < |\mathbf{g}_i| \leq \frac{\|\mathbf{g}\|_\infty}{16}\}$. By the assumption of the theorem, $\mathbb{P}(E_i) \geq \nu$, and on $E_i$ we have both $Q(\mathbf{v}_i) = 0$ and

$$\frac{|\mathbf{g}_i| - \|\mathbf{g}\|_\infty/127}{\epsilon} \geq \frac{\|\mathbf{g}\|_\infty/60 - \|\mathbf{g}\|_\infty/127}{\epsilon}$$

$$\geq \frac{\|\mathbf{g}\|_\infty}{128\epsilon}.$$

Since $\|\mathbf{g}\|_\infty \geq g_\infty$ almost surely and $g_\infty \geq 256\epsilon$,

$$\frac{\|\mathbf{g}\|_\infty}{128\epsilon} - 1 \geq \frac{g_\infty}{128\epsilon} - 1$$
$$\geq \frac{1}{2} \cdot \frac{g_\infty}{128\epsilon} = \frac{g_\infty}{256\epsilon}.$$

Set $\tau_0 := \frac{g_\infty}{256\epsilon}$. Then, for $t = \tau_0$,

$$\mathbb{P}\left( \left| \frac{\mathbf{m}_i}{\sqrt{\mathbf{v}_i} + \epsilon} - \frac{Q(\mathbf{m}_i)}{\sqrt{Q(\mathbf{v}_i)} + \epsilon} \right| \geq t \right) \geq \mathbb{P}(E_i)$$
$$\geq \nu.$$

Therefore, using $\mathbb{E}[X^2] \geq t^2 \mathbb{P}(X \geq t)$ for nonnegative $X$,

$$\mathbb{E}\left[ \left( \frac{\mathbf{m}_i}{\sqrt{\mathbf{v}_i} + \epsilon} - \frac{Q(\mathbf{m}_i)}{\sqrt{Q(\mathbf{v}_i)} + \epsilon} \right)^2 \right] \geq \nu\, \tau_0^2 = \nu \frac{g_\infty^2}{(256\epsilon)^2}.$$

Summing over $i = 1, \ldots, d$ and using $\boldsymbol{\theta}^{(1)} - \tilde{\boldsymbol{\theta}}^{(1)} = \alpha\left( \frac{\mathbf{m}}{\sqrt{\mathbf{v}} + \epsilon} - \frac{Q(\mathbf{m})}{\sqrt{Q(\mathbf{v})} + \epsilon} \right)$,

$$\mathbb{E}\|\boldsymbol{\theta}^{(1)} - \tilde{\boldsymbol{\theta}}^{(1)}\|_2^2 \geq \alpha^2 d\nu \frac{g_\infty^2}{(256\epsilon)^2}.$$

∎

### C.2.2 PROOF OF THEOREM 2

**Proof** We compare the two updates and isolate the effect of quantizing the momentum. The gradient terms cancel, leaving

$$\tilde{\boldsymbol{\theta}}^{(t+1)} - \boldsymbol{\theta}^{(t+1)} = -\eta(\mathbf{g}^{(t)} + \rho Q(\mathbf{m}^{(t)})) + \eta(\mathbf{g}^{(t)} + \rho \mathbf{m}^{(t)})$$
$$= -\eta\rho \left( Q(\mathbf{m}^{(t)}) - \mathbf{m}^{(t)} \right).$$

Taking squared norms and expanding coordinatewise yields

$$\|\tilde{\boldsymbol{\theta}}^{(t+1)} - \boldsymbol{\theta}^{(t+1)}\|_2^2 = \eta^2 \rho^2 \sum_{i=1}^d \left( Q(\mathbf{m}^{(t)})_i - \mathbf{m}_i^{(t)} \right)^2.$$

By Definition 1, each coordinate is perturbed by at most $\|\mathbf{m}^{(t)}\|_\infty / 127$, i.e., $|Q(\mathbf{m}^{(t)})_i - \mathbf{m}_i^{(t)}| \leq \|\mathbf{m}^{(t)}\|_\infty / 127$. Applying this inside the sum gives

$$\|\tilde{\boldsymbol{\theta}}^{(t+1)} - \boldsymbol{\theta}^{(t+1)}\|_2^2 \leq \eta^2 \rho^2 \sum_{i=1}^d \left( \|\mathbf{m}^{(t)}\|_\infty / 127 \right)^2$$
$$= d\, \eta^2 \rho^2 \left( \|\mathbf{m}^{(t)}\|_\infty / 127 \right)^2,$$

which is the claimed bound. ∎

### C.2.3 PROOF OF THEOREM 3

**Proof** From the two updates,

$$\widetilde{\mathbf{W}}^{(t)} - \mathbf{W}^{(t)} = -\alpha\left( \widetilde{\mathbf{O}}^{(t)} - \mathbf{O}^{(t)} \right) \quad \Rightarrow \quad \|\widetilde{\mathbf{W}}^{(t)} - \mathbf{W}^{(t)}\|_F = \alpha \|\widetilde{\mathbf{O}}^{(t)} - \mathbf{O}^{(t)}\|_F.$$

The momentum matrices $\mathbf{M}^{(t)}$ and $\widetilde{\mathbf{M}}^{(t)}$ are assumed to be full column rank with $\sigma_{\min}(\mathbf{M}^{(t)}), \sigma_{\min}(\widetilde{\mathbf{M}}^{(t)}) \geq s > 0$. For full-column-rank matrices, the (rectangular) polar-factor map satisfies

$$\|\widetilde{\mathbf{O}}^{(t)} - \mathbf{O}^{(t)}\|_{\mathrm{F}} \ \leq \ \frac{2}{\sigma_{\min}(\mathbf{M}^{(t)}) + \sigma_{\min}(\widetilde{\mathbf{M}}^{(t)})} \|\widetilde{\mathbf{M}}^{(t)} - \mathbf{M}^{(t)}\|_{\mathrm{F}} \ \leq \ \frac{1}{s} \|\widetilde{\mathbf{M}}^{(t)} - \mathbf{M}^{(t)}\|_{\mathrm{F}},$$

(see (Bhatia, 2013, Thm. VII.5.1(a)) and its extension to full column rank in (Li, 1995, Thm. 2)). Hence

$$\|\widetilde{\mathbf{W}}^{(t)} - \mathbf{W}^{(t)}\|_{\mathrm{F}} \ \leq \ \alpha \frac{1}{s} \|\widetilde{\mathbf{M}}^{(t)} - \mathbf{M}^{(t)}\|_{\mathrm{F}} \ = \ \alpha \frac{\beta}{s} \|Q(\mathbf{M}^{(t-1)}) - \mathbf{M}^{(t-1)}\|_{\mathrm{F}}.$$

By Definition 1, each entry changes by at most $\|\operatorname{vec}(\mathbf{M}^{(t-1)})\|_\infty/127$, so $\|Q(\mathbf{M}^{(t-1)}) - \mathbf{M}^{(t-1)}\|_{\mathrm{F}}^2 \leq d\big(\|\operatorname{vec}(\mathbf{M}^{(t-1)})\|_\infty/127\big)^2$. Squaring both sides completes the proof. ∎

## C.3 ADAM ALGORITHM

For completeness, we reproduce the base Adam algorithm below (Algorithm 1 in Kingma (2014)).

---
**Algorithm 4** Adam (using $\boldsymbol{\theta}, \mathbf{g}, \mathbf{m}, \mathbf{v}$)
---
1: **Input:** step size $\alpha$, decay rates $\beta_1, \beta_2 \in [0, 1)$, $\epsilon > 0$
2: **Initialize:** $\boldsymbol{\theta}^{(0)}$, $\tilde{\mathbf{m}}^{(0)} = \mathbf{0}$, $\tilde{\mathbf{v}}^{(0)} = \mathbf{0}$
3: **for** $t = 1, 2, \dots$ **do**
4: $\qquad \mathbf{g}^{(t)} \leftarrow \nabla f_t(\boldsymbol{\theta}^{(t-1)})$
5: $\qquad \tilde{\mathbf{m}}^{(t)} \leftarrow \beta_1 \tilde{\mathbf{m}}^{(t-1)} + (1 - \beta_1)\mathbf{g}^{(t)}$
6: $\qquad \tilde{\mathbf{v}}^{(t)} \leftarrow \beta_2 \tilde{\mathbf{v}}^{(t-1)} + (1 - \beta_2)(\mathbf{g}^{(t)})^2$
7: $\qquad \mathbf{m}^{(t)} \leftarrow \tilde{\mathbf{m}}^{(t)}/(1 - \beta_1^t)$
8: $\qquad \mathbf{v}^{(t)} \leftarrow \tilde{\mathbf{v}}^{(t)}/(1 - \beta_2^t)$
9: $\qquad \boldsymbol{\theta}^{(t)} \leftarrow \boldsymbol{\theta}^{(t-1)} - \alpha \mathbf{m}^{(t)}/(\sqrt{\mathbf{v}^{(t)}} + \epsilon)$
10: **end for**

---

## C.4 EXACT MUON OPTIMIZER

For completeness, we provide the update formula for the exact Muon algorithm (without weight decay) here. As opposed to eqn. (1), this formula uses the exact polar factor $\mathbf{O} = \mathbf{U}^{(t)}\mathbf{V}^{(t)\top}$

$$
\begin{aligned}
\mathbf{M}^{(t)} &:= \beta \mathbf{M}^{(t-1)} + \mathbf{G}_t, \\
\mathbf{M}^{(t)} &= \mathbf{U}^{(t)}\mathbf{S}^{(t)}\mathbf{V}^{(t)\top} \text{ (thin SVD)}, \\
\mathbf{O}^{(t)} &:= \mathbf{U}^{(t)}\mathbf{V}^{(t)\top}, \\
\mathbf{W}^{(t)} &= \mathbf{W}^{(t-1)} - \alpha \mathbf{O}^{(t)}.
\end{aligned}
\tag{4}
$$

In the quantized variant, only the previous momentum is quantized and then de-quantized, leading to the following updates where $Q$ is defined in Definition 1:

$$
\begin{aligned}
\widetilde{\mathbf{M}}^{(t)} &:= \beta Q(\mathbf{M}^{(t-1)}) + \mathbf{G}_t, \\
\widetilde{\mathbf{M}}^{(t)} &= \widetilde{\mathbf{U}}^{(t)}\widetilde{\mathbf{S}}^{(t)}\widetilde{\mathbf{V}}^{(t)\top} \text{ (thin SVD)}, \\
\widetilde{\mathbf{O}}^{(t)} &:= \widetilde{\mathbf{U}}^{(t)}\widetilde{\mathbf{V}}^{(t)\top}, \\
\widetilde{\mathbf{W}}^{(t)} &= \mathbf{W}^{(t-1)} - \alpha \widetilde{\mathbf{O}}^{(t)}.
\end{aligned}
\tag{5}
$$

## C.5 EMPIRICAL VALIDATION

In this section, we provide empirical validation for the theorems of Section 4.1 using a simple experimental setup. We train a two-layer fully-connected network with one hidden layer of width

256 and ReLU activation, mapping $28 \times 28$ pixel inputs to 10 logits. Inputs are normalized with the standard MNIST mean and variance $(0.1307, 0.3081)$, and we use the standard training split from `torchvision` with a batch size of 128 and cross-entropy loss.

**Theorem 1** - We measure that approximately 29% of gradient entries in the loss gradient, $\mathbf{g}$, fall within the range $[\|\mathbf{g}\|_\infty/60, \|\mathbf{g}\|_\infty/16]$ from a Kaiming Uniform initialization of the model weights. This indicates that the gradient coordinate assumption of Theorem 1 holds in practice, and hence provides a realistic explanation for the divergence observed after a single step of Adam.

**Theorem 2** - We measure the quantization error $\|\tilde{\boldsymbol{\theta}}^{(t+1)} - \boldsymbol{\theta}^{(t+1)}\|_2^2$ as a proportion of the error bound in Theorem 2 for the SGD+M optimizer with learning rate $0.1$ and momentum $0.9$. For all $t$ over $1400$ training steps, the ratio of the true squared $\ell_2$ quantization error and the error bound stays between $0.06$ and $0.085$. By taking the square root of both sides in the error bound, we see that the true $\ell_2$-norm quantization error is accurately predicted by the theoretical bound up to a small constant factor.

**Theorem 3** - We measure the quantization error $\|\widetilde{\mathbf{W}}^{(t)} - \mathbf{W}^{(t)}\|_F^2$ of the first layer weight matrix for the Muon optimizer with $\alpha = 0.2$ and $\beta = 0.95$. We observe that the ratio of the true squared Frobenius-norm quantization error to the bound of Theorem 3 stays between $2 \times 10^{-6}$ and $1.4 \times 10^{-5}$. The gap is explained by the fact that the proof of our theorem depends on the weak assumption that original and quantized matrices have full column rank $r$ with the $r$-th singular value of each matrix lower bounded by $s$. In actuality, this condition is pessimistic, as the perturbation of the weight matrix is not aligned with the singular vector associated with the $r$-th singular value. More complex assumptions on the spectrum of $\mathbf{W}$, paired with probabilistic assumptions on the quantization noise and a polar-factor perturbation analysis that accounts for additional spectral information (see Li & Sun (2003), for example), would produce a tighter bound. However, this analysis is outside the scope of this work.

