# OpenReview forum: "On quantizing the state of the Muon optimizer"
_ICLR.cc/2026/Conference — Submitted to ICLR 2026_

### Official Review · Reviewer_1fcx · 2025-10-23

**Soundness:** 2
**Presentation:** 3
**Contribution:** 1
**Rating:** 2
**Confidence:** 5

**Summary:**

This work studies why linear quantization is unstable for Adam but works well for SGD and Muon from a theoretical perspective. The authors evalute 8-bit Muon with both linear and dynamic quantization on pretraining (1.6B model on 4B tokens) and finetuning task.

**Strengths:**

The writing is clear.

**Weaknesses:**

- The main weakness is that the core part of this paper focuses on why low-bit Adam becomes unstable under linear quantization, whereas SGD and Muon do not (Section 4), which is less relevant to the title. Furthermore, the authors do not propose any new technique for 8-bit Muon. It appears that 8-bit Muon faces no particular challenges, and existing techniques already work well.
- The experimental setup is limited. Although the 1.6 B scale is reasonable for an academic paper, the tokens-per-parameter ratio (4 / 1.6 = 2.5) is too small.
- The assumption in Theorem 1, gradient coordinates of moderate size occur with non-negligible probability, is not justified.
- Theorems 1, 2, and 3 lack empirical justification. It would be better to measure the quantization error under linear quantization and compare it with these theoretical bounds.

**Questions:**

See weakness above.

---

> ### Author Response · Authors · 2025-11-28
>
> We thank the reviewer for taking the time to review our paper, and for their valuable comments
>
> Responses to weaknesses:
>
> > **Existing techniques already work well.**
>
>
> We believe that our findings that simple linear quantization works well for Muon and SGD are novel and cannot be found in literature. Before our paper, the common flavor of quantization was dynamic quantization. Previously, **Dettmers et al 2021** empirically showed that linear quantization does not work well for AdamW, but did not go beyond this. We are the first to theoretically prove the drawbacks of using linear quantization for AdamW. **We also systematically tested variants of quantized Muon by conducting chinchilla-optimal experiments (see Table 1 and Table 2)** and demonstrated clearly that Muon state quantization leads to almost no degradation in validation loss while quality on benchmarks is similar or unchanged. We do acknowledge our title is a bit narrow, and we will update this in the draft. Thanks for the feedback.
>
> > **The tokens-per-parameter ratio (4 / 1.6 = 2.5) is too small**
>
> You are right to point this out, thank you for pushing us to do this. **We re-ran the experiments using Chinchilla-optimal ratios of 20 tokens to each parameter**. The results are in Table 1 and Table 2 below. From the tables it is clear that **(a) all model sizes have a small validation loss gap between the quantized and the unquantized versions and (b) there is no appreciable loss in quality when tested on 6 different benchmarks for the XL model, and there is a small drop for medium and large models.**
>
> > **The assumption in Theorem 1, gradient coordinates of moderate size occur with non-negligible probability, is not justified.**
>
> We have added an experiment in the appendix that logs the distribution of gradient coordinate magnitudes from a Kaiming Uniform initialization. Approximately 29% of coordinates fall within the range assumed in Theorem 1, supporting that the assumption holds in practice (see Appendix C.5 for details).
>
>
> > **Theorems 1, 2, and 3 lack empirical justification. It would be better to measure the quantization error under linear quantization and compare it with these theoretical bounds.**
>
>
> **We have additional experiments that measure the linear quantization error across training steps for SGD+M and Muon and compare it to our theoretical bounds. In all cases, the observed errors remain below the predicted bounds, empirically supporting Theorems 2 and 3 (see Appendix C.5)**. For Theorem 1, we already showed (line 264) that AdamW diverges almost immediately under the same setup, which is consistent with its theoretical behavior.
>
> **Table 1 – validation loss**
>
> | Model | Muon (full) | Muon‑8bit‑linear | Abs. Δ | Rel. Δ (%) |
> | --- | --- | --- | --- | --- |
> | Medium | 2.78779 | 2.78865 | 0.00086 | 0.03 |
> | Large | 2.58743 | 2.59320 | 0.00577 | 0.22 |
> | XL | 2.42661 | 2.42747 | 0.00086 | 0.04 |
>
> Validation loss comparison of full‑precision Muon vs. 8‑bit linear Muon across model sizes.
>
> ---
>
> **Table 2 – LM-Eval accuracies (higher is better)**
>
> | Task | Medium Muon | Medium Muon‑8bit‑lin | Large Muon | Large Muon‑8bit‑lin | XL Muon | XL Muon‑8bit‑lin |
> | --- | --- | --- | --- | --- | --- | --- |
> | ARC‑Challenge | 0.273 | 0.269 | 0.294 | 0.281 | 0.324 | 0.329 |
> | ARC‑Easy | 0.473 | 0.465 | 0.527 | 0.514 | 0.603 | 0.591 |
> | BoolQ | 0.616 | 0.573 | 0.602 | 0.591 | 0.586 | 0.600 |
> | HellaSwag | 0.343 | 0.344 | 0.399 | 0.399 | 0.502 | 0.500 |
> | LAMBADA | 0.236 | 0.227 | 0.312 | 0.299 | 0.387 | 0.384 |
> | MMLU | 0.230 | 0.230 | 0.237 | 0.233 | 0.247 | 0.249 |
> | **Avg. (6 tasks)** | **0.362** | **0.351** | **0.395** | **0.386** | **0.441** | **0.442** |

---

### Official Review · Reviewer_cc14 · 2025-10-25

**Soundness:** 2
**Presentation:** 3
**Contribution:** 2
**Rating:** 6
**Confidence:** 4

**Summary:**

This paper investigates applying quantization to the Muon optimizer state, in a manner similar to the prior work quantizing AdamW state by Dettmers et al.  The paper argues that, in AdamW, the key source of error for simple linear quantization schemes is via error in the second-order velocity term.  Since Muon (like SGD+M) does not use such a term, it trains fine with simple linear quantization.  But it also trains fine with dynamic quantization.  Either way it saves a lot of space compared to vanilla Muon.

**Strengths:**

Paper was fairly clear, easy to read, well-organized.

The key idea, that Muon is easier to quantize because it doesn't share AdamW's velocity term, is interesting.

**Weaknesses:**

Overall:
- The main overall idea, to quantize the Muon optimizer states, was a fairly obvious thing to try, although someone has to do it and test it.  So the main value of a paper trying something obvious like that is to do a really thorough empirical study/investigation.
- I didn't find the experimental work that comprehensive or perfectly done.  I felt there was more empirical work that could have been done to firmly establish the main premise (that velocity is the culprit for AdamW not doing well with linear).  I mean, creative ways to assess this beyond what's in the paper now (and beyond what I can think of).  I felt the key theoretical aspects of this could have been in the main paper rather than the appendix.
- So I feel it's kind of a borderline paper.

In more detail:

“Several large-scale studies have confirmed Muon’s ability to achieve a 2x efficiency… This includes extremely large models up to a trillion parameters in size”
- Really? Did they confirm at 1T size that Muon has 2x efficiency over AdamW (like, actually ablating the optimizer)?  Maybe this is a bit misleading.  Especially given my next point.

This recent paper argues: “the speedup of matrix-based optimizers is inversely proportional to model scale, decreasing from 1.4× over AdamW for 0.1B parameter models to merely 1.1× for 1.2B parameter models.”
- Wen - Fantastic PT optimizers and where to find them - 2509.02046v2
- I'm just saying, let's acknowledge that Muon > AdamW isn't established at scale.

It seems like the fact that the error in Theorem 1 arising primarily from error in the second moment is a key part of the paper, yet it’s in the appendix!

- Since there is quite different behaviour of optimizers between language and image tasks (e.g., https://arxiv.org/abs/2304.13960), I don’t understand why we don’t confirm the lack of divergence for SGD+M on language tasks?  Like, it seems like an unnecessary leap to say, “Since SGD works well with linear quantization,” with the implication this holds in LLMs.

I have some issues with the experimental setup:
- ROPE is known to have some precision issues, I wonder how that affects things versus using AliBi
- Only the smallest model is trained to compute-optimal (20 tokens-per-parameter), so it's not clear that many of these settings are practically relevant.
- Using 0.9 and 0.999 Betas for AdamW doesn’t seem quite standard to me.  It seems especially important to test different beta2, because I would think beta2 would affect the second-moment error, right?
- Using the same LR for 3 different model sizes doesn’t seem optimal
- Using the same batch size across all scales also doesn’t seem quite right

- “Since most of the peak learning rates have been tuned for AdamW” – I don’t really understand what this is referring to.  Tuned by whom?

Nitpicks:
- Equation (2), last line, λ should multiply by W^(t-1), not W^(t).
- Section 4.3 could benefit from a concise statement like, “because Muon, like SGD, doesn’t use a second-order momentum term, we hypothesize it will train stably with linear quantization.”

**Questions:**

- What’s the real significance of the finding that Muon works well with linear quantization?  Like, we say that AdamW needs “careful blockwise + dynamic quantization,” implying that it takes more work than a simple linear scheme.  Is that so?  Are there other drawbacks of dynamic?

- What are the drawbacks of quantization in general?  It saves memory, but are there not other tradeoffs?

---

> ### Author Response · Authors · 2025-11-28
>
> We really appreciate the reviewer spending time to provide detailed comments and feedback. Here are our responses:
>
> > **Experimental work should be more comprehensive**
>
> Based on the feedback, we re-ran the experiments using Chinchilla-optimal ratios of 20 tokens to each parameter. We also tested the models on 6 different benchmarks. The results are in Table 1 and Table 2 below. From the tables it is clear that **(a) all model sizes have a small validation loss gap between the quantized and the unquantized versions and (b) there is no appreciable loss in quality when tested on 6 different benchmarks for the XL model, and there is a small drop for medium and large models.**
>
> > **About Muon’s scalability and 2x efficiency**
>
> We acknowledge that we didn’t want the purported efficiency and scalability to be connected. The second sentence was meant to talk about scalability, and not efficiency found in other papers. We have fixed this in the new draft
>
> > **It seems like the fact that the error in Theorem 1 arising primarily from error in the second moment is a key part of the paper, yet it’s in the appendix!**
>
> Note, we do mention this on line 210 of the main text, but we agree further emphasis is needed. We have added the clarification you recommended to Section 4.3
>
>
> Responses to experimental questions
>
> > **ROPE is known to have some precision issues, I wonder how that affects things versus using AliBi**
>
> We agree this could be a useful study. Since our experiments focus on model sizes and optimizers, we will leave this for future work
>
> > **Only the smallest model is trained to compute-optimal (20 tokens-per-parameter), so it's not clear that many of these settings are practically relevant.**
>
> You are 100% right to call this out. Based on your feedback and that of other reviewers, we trained the models using a compute-optimal (20 tokens per parameter) setup. Results can be found in Table 1 and Table 2
>
> > **Using 0.9 and 0.999 Betas for AdamW doesn’t seem quite standard to me. It seems especially important to test different beta2, because I would think beta2 would affect the second-moment error, right?**
>
> Most of our experiments are focused on Muon vs Muon 8-bit. For AdamW, we re-ran with beta2 = 0.95 and that didn’t materially change the results
>
> > **Using the same LR for 3 different model sizes doesn’t seem optimal**
>
> We have now re-run the experiments with the size optimal LRs from the GPT3 paper and updated the draft. There is no obvious change to our results and conclusions
>
> > **Using the same batch size across all scales also doesn’t seem quite right**
>
> We have fixed this, see the draft. There is no obvious change to our results and conclusions
>
> > **“Since most of the peak learning rates have been tuned for AdamW” – I don’t really understand what this is referring to. Tuned by whom?**
>
> Thanks for the comment. We have clarified in the draft that these are well-tuned by the community and OpenAI for GPT3 for the AdamW optimizer, and we use the same values for Muon and Muon 8-bit.
>
> >**Equation (2), last line, λ should multiply by W^(t-1), not W^(t).**
>
> Thanks for pointing out this typo. We have fixed this
>
> > **Section 4.3 could benefit from a concise statement like, “because Muon, like SGD, doesn’t use a second-order momentum term, we hypothesize it will train stably with linear quantization.”**
>
> You are right. We have added this line to the draft. Thanks for the great input.
>
> Responses to questions
>
> > **Significance of findings that Muon works well with linear quantization and AdamW needs “careful blockwise + dynamic quantization,”**
>
> Indeed, AdamW doesn’t work well with linear quantization. We comprehensively validate that linear quantization works for Muon and SGD while providing theoretical insights, there is currently no established literature on this topic. Furthermore, to the best of our knowledge, this is the first work to provide a theoretical explanation for why Adam is unstable under linear quantization.
>
> The biggest significance is that for non-Adam optimizers, linear becomes a viable option. It is simple to implement and understand, when compared to dynamic quantization. Contrary to popular perception, many practitioners can probably directly use linear quantization for SGD and Muon like optimizers without worrying about the complexity of dynamic quantization
>
>
> > **Drawbacks for quantization**
>
> In table 2, we see a small drop for smaller sizes for various benchmarks. This gap seems to be mitigated at the XL size. Based on findings from papers like GPTQ, which study quantization of weights, it looks like larger models can tolerate quantization better.
>
> (Tables 1 and 2 are in the second response)

---

> > ### Author Response · Authors · 2025-11-28
> >
> > **Table 1 – validation loss**
> >
> > | Model | Muon (full) | Muon‑8bit‑linear | Abs. Δ | Rel. Δ (%) |
> > | --- | --- | --- | --- | --- |
> > | Medium | 2.78779 | 2.78865 | 0.00086 | 0.03 |
> > | Large | 2.58743 | 2.59320 | 0.00577 | 0.22 |
> > | XL | 2.42661 | 2.42747 | 0.00086 | 0.04 |
> >
> > Validation loss comparison of full‑precision Muon vs. 8‑bit linear Muon across model sizes.
> >
> > ---
> >
> > **Table 2 – LM-Eval accuracies (higher is better)**
> >
> > | Task | Medium Muon | Medium Muon‑8bit‑lin | Large Muon | Large Muon‑8bit‑lin | XL Muon | XL Muon‑8bit‑lin |
> > | --- | --- | --- | --- | --- | --- | --- |
> > | ARC‑Challenge | 0.273 | 0.269 | 0.294 | 0.281 | 0.324 | 0.329 |
> > | ARC‑Easy | 0.473 | 0.465 | 0.527 | 0.514 | 0.603 | 0.591 |
> > | BoolQ | 0.616 | 0.573 | 0.602 | 0.591 | 0.586 | 0.600 |
> > | HellaSwag | 0.343 | 0.344 | 0.399 | 0.399 | 0.502 | 0.500 |
> > | LAMBADA | 0.236 | 0.227 | 0.312 | 0.299 | 0.387 | 0.384 |
> > | MMLU | 0.230 | 0.230 | 0.237 | 0.233 | 0.247 | 0.249 |
> > | **Avg. (6 tasks)** | **0.362** | **0.351** | **0.395** | **0.386** | **0.441** | **0.442** |

---

### Official Review · Reviewer_v57K · 2025-11-03

**Soundness:** 1
**Presentation:** 2
**Contribution:** 1
**Rating:** 2
**Confidence:** 3

**Summary:**

The paper shows the theoretical analysis of one step error in low precision Adam update, SGD update and Muon update. By showing the theoretical bound, the authors claim that Muon has small error similar to SGD, much smaller than Adam. The experimental results show the 8-bit Muon can perform well in pretraining setting and finetuning setting.

**Strengths:**

1. The paper provides error analysis on one step update for Adam, SGD  and Muon, roughly showing the error of each algorithm.

2. The paper shows experimental results that 8-bit Muon perform well under 1.6B model with 2B tokens pretrain and 3B model SFT.

**Weaknesses:**

1.  All of the error analysis should go through the whole training process instead of one-step analysis.

2. The error analysis of Adam should related to $\beta_1$ and $\beta_2$.

3. The NS iteration will not exact give the $O$ matrix for Muon. But the analysis is mainly based on $O$ matrix.

4. It is not enough for training 1.6B model with 2B tokens.

**Questions:**

See Weaknesses.

---

> ### Author Response · Authors · 2025-11-28
>
> We are thankful to the reviewer for their valuable comments and insights
>
> > **the error analysis should go through the whole training process**
>
> We believe a single-step error analysis is sufficient to explain the phenomena we study. Empirically, Adam diverges after just one step, which is consistent with Theorem 1. In contrast, both our theory and experiments show that the single-step error for Muon and GD is much smaller, which already accounts for their stable behavior throughout training.
> Could you clarify what specific insights a multi-step analysis would add beyond this? From our perspective, the single-step characterization already matches and explains the optimizer dynamics observed empirically.
>
> > **The error analysis of Adam should related to  beta1 and beta2.**
>
> We respectfully disagree with this point. These parameters do not influence the stability of the first Adam step. As noted earlier, even the first step of Adam already diverges, so relying on those specific values would, in fact, lead to a strictly weaker statement.
>
> > **The NS iteration will not exact give the  matrix for Muon.**
>
> We conduct our analysis with respect to the exact polar factor—the target of NS-type routines—in order to cleanly separate optimizer behavior from inner-loop approximation effects. A comprehensive numerical study of NS truncation error is beyond the scope of this paper. Moreover, formulating the theorem in terms of the exact polar factor makes it directly applicable to recent methods that approximate it in different ways (e.g., https://arxiv.org/abs/2506.10935)
>
> > **It is not enough for training 1.6B model with 2B tokens.**
>
> We appreciate the suggestion to revisit our experiments. In response, **we have re-run all experiments using Chinchilla-optimal training**; the updated results are presented in Table 1, and the corresponding benchmark performance is shown in Table 2. The findings clearly support our original claims: **the validation loss gap is very small for all model sizes, and the gap on downstream benchmarks remains uniformly negligible for the XL model, and is small for medium/large models.**
>
> **Table 1 – validation loss**
>
> | Model | Muon (full) | Muon‑8bit‑linear | Abs. Δ | Rel. Δ (%) |
> | --- | --- | --- | --- | --- |
> | Medium | 2.78779 | 2.78865 | 0.00086 | 0.03 |
> | Large | 2.58743 | 2.59320 | 0.00577 | 0.22 |
> | XL | 2.42661 | 2.42747 | 0.00086 | 0.04 |
>
> Validation loss comparison of full‑precision Muon vs. 8‑bit linear Muon across model sizes.
>
> ---
>
> **Table 2 – LM-Eval accuracies (higher is better)**
>
> | Task | Medium Muon | Medium Muon‑8bit‑lin | Large Muon | Large Muon‑8bit‑lin | XL Muon | XL Muon‑8bit‑lin |
> | --- | --- | --- | --- | --- | --- | --- |
> | ARC‑Challenge | 0.273 | 0.269 | 0.294 | 0.281 | 0.324 | 0.329 |
> | ARC‑Easy | 0.473 | 0.465 | 0.527 | 0.514 | 0.603 | 0.591 |
> | BoolQ | 0.616 | 0.573 | 0.602 | 0.591 | 0.586 | 0.600 |
> | HellaSwag | 0.343 | 0.344 | 0.399 | 0.399 | 0.502 | 0.500 |
> | LAMBADA | 0.236 | 0.227 | 0.312 | 0.299 | 0.387 | 0.384 |
> | MMLU | 0.230 | 0.230 | 0.237 | 0.233 | 0.247 | 0.249 |
> | **Avg. (6 tasks)** | **0.362** | **0.351** | **0.395** | **0.386** | **0.441** | **0.442** |

---

### Official Review · Reviewer_K3yi · 2025-11-03

**Soundness:** 2
**Presentation:** 3
**Contribution:** 2
**Rating:** 4
**Confidence:** 4

**Summary:**

This paper introduces 8-bit variants of the Muon optimizer using blockwise quantization, supporting both linear and dynamic quantization schemes. The authors provide theoretical analysis explaining why Muon remains stable under linear quantization (unlike AdamW): Theorem 1 shows AdamW's instability stems from quantizing the second-moment denominator, while Theorems 2-3 provide uniform error bounds for SGD and Muon under quantization. Experimentally, the paper demonstrates that 8-bit Muon achieves ~74% memory reduction versus full-precision Muon while maintaining competitive performance on pretraining (GPT-style models up to 1.6B parameters on FineWeb) and finetuning (Llama 3.2 3B on Tulu-3). The work validates that Muon's orthogonalization-based updates are inherently more robust to quantization than adaptive methods.

**Strengths:**

-- Originality: The paper provides novel theoretical analysis (Theorems 1-3) explaining optimizer robustness to quantization. While the quantization technique itself isn't new, identifying the structural property that makes Muon robust (absence of second-moment division) is an interesting insight. The systematic comparison between linear and dynamic quantization schemes for Muon is also original.

-- Quality: The experimental work is thorough within its scope. Multiple model sizes (97M-1.6B), proper baselines (AdamW-32, AdamW-8D, Muon-32), ablation studies, and stability analyses demonstrate careful execution. The theoretical proofs are rigorous. The ImageNet experiment elegantly isolates the core phenomenon. Training details are comprehensive for reproducibility.

-- Clarity: The paper is well-structured with clear writing. The motivation (LLM memory constraints) is compelling and immediately understandable. The theoretical sections balance rigor with accessibility. Figures effectively communicate key results (Figure 1 for memory, Figure 4 for ImageNet).

**Weaknesses:**

-- Limited novelty and scope: The core technique (blockwise quantization) is from prior work; this paper applies it to Muon. While the theoretical analysis is valuable, the contribution feels incremental—demonstrating that an existing technique works for a new optimizer. The focus on a single optimizer (Muon) without broader comparative analysis limits the contribution's scope.

-- Insufficient scale validation: Evaluation stops at 3B parameters, well below modern LLM scales (7B-70B+). Given that memory constraints are most acute for larger models, the limited scale raises questions about practical applicability. Do the 1-2% performance gaps compound at larger scales? Does the quantization remain stable? These critical questions are unanswered.

-- Missing comparisons: The paper doesn't compare with other memory-efficient training methods (Adafactor, GaLore, LoRA, Adam-mini) that address the same memory constraints through different mechanisms. Understanding the tradeoffs between 8-bit Muon and these alternatives is essential for practitioners but absent from the paper.

-- Performance degradation not fully addressed: The consistent 1-2% performance drop on smaller models (XS/Small/Medium in Table 3) is acknowledged but not deeply analyzed. For practitioners, especially at scale, these gaps matter. When/why do they occur? Can they be mitigated? Are they acceptable tradeoffs? Deeper analysis would strengthen the work.

**Questions:**

-- Scalability to larger models: Have you conducted any preliminary experiments beyond 3B parameters? What specific challenges do you anticipate at 7B, 13B, or 70B scales? Given that memory constraints are most acute for large models, this seems critical to validate.


-- Theoretical quantities in practice: Can you provide empirical measurements of the minimum singular values (from Theorem 3) and momentum norms (from Theorem 2) during actual training runs? Do these quantities remain in regimes where your bounds are meaningful?


-- Performance gap analysis: For the 1-2% performance drops on XS/Small/Medium models (Table 3), can you identify when/why these occur? Have you attempted hyperparameter adjustment or longer training to close this gap? Do the gaps persist or widen at larger scales?

---

> ### Author Response · Authors · 2025-11-28
>
> We thank the reviewer for their detailed insights and valuable comments.
>
> Response to weaknesses:
>
> > **Limited novelty and scope**
>
> Thank you for recognizing the value of our theoretical contribution. We would like to emphasize that comprehensively validating that linear quantization works for Muon and SGD while providing theoretical insight is non-trivial, and there is currently no established literature on this topic. Moreover, **our experiments are now Chinchilla-optimal (see below)** and **show that even simple linear quantization yields models whose quality is close to or indistinguishable from their full-precision counterparts**. Furthermore, to the best of our knowledge, this is the first work to provide a theoretical explanation for why Adam is unstable under linear quantization.
>
> > **Insufficient scale validation**
>
> Thank you for raising this concern. Due to a limited compute budget, we are constrained in training models larger than 3B+ parameters. Nevertheless, **our Chinchilla-optimal results indicate that (a) the loss gap is very small for all model sizes, and (b) performance on downstream benchmarks is unchanged for the XL model and very similar for the medium and large models, under quantization**. In addition, our theoretical results establish that, for one optimization step, the discrepancy between the quantized and full-precision updates is bounded for both SGD and Muon. Taken together, **these findings give us strong evidence that at larger scales our method would remain stable.**
>
> > **Missing comparisons**
>
> All the methods pointed out by the reviewer do not operate on the optimizer state directly. Our method can easily be combined with methods like GaLore and LORA. Thus, we did not feel the need to run such ablations
>
> > **Performance degradation not fully addressed**
>
> Thank you for pushing us to improve here. You are 100% right, our experiments had room for improvement. Based on your feedback, we re-ran the experiments at chinchill optimal scale of 20 tokens per parameter for all model sizes. This resulted in a **clear trend of the validation loss gap being small across all model sizes (Table 1)**. Benchmarks reveal that the gap between quantized and full-precision models is small for medium/large and zero for XL (Table 2). We really appreciate your insights here!
>
> Response to questions:
>
> > **Scalability to larger models**
>
> As mentioned above, our compute budget is limited, making it difficult for us to train 3B+ models. At the same time, distributing Muon is non-trivial and requires substantial engineering changes and insights (e.g., see [arXiv:2504.05295]). This is precisely why we focused on a simple scheme such as linear quantization. Scaling to 3B+ models would require moving beyond standard distributed data parallelism, which we view as important future work. We are happy to explicitly state this as a limitation and future direction in the paper.
>
> > **Theoretical quantities in practice**
>
> **We have added experiments in Appendix C.5  to empirically validate Theorems 1–3**. For Theorem 1, the assumption on gradient-coordinate magnitudes is supported in practice: under a Kaiming Uniform initialization, about 29% of coordinates fall in the assumed range. For Theorems 2 and 3, we log the theoretical bounds and the observed quantization error across training steps for SGD+M and Muon, and we find that our theory accurately predicts observed behavior.
>
>
> > **Performance gap analysis**
>
> As discussed above, the validation loss gap closes with model size. Also, benchmarks reveal no gap between optimizers for XL, and a small drop for medium and large. See tables 1 and 2 below
>
> **Table 1 – validation loss**
>
> | Model | Muon (full) | Muon‑8bit‑linear | Abs. Δ | Rel. Δ (%) |
> | --- | --- | --- | --- | --- |
> | Medium | 2.78779 | 2.78865 | 0.00086 | 0.03 |
> | Large | 2.58743 | 2.59320 | 0.00577 | 0.22 |
> | XL | 2.42661 | 2.42747 | 0.00086 | 0.04 |
>
> Validation loss comparison of full‑precision Muon vs. 8‑bit linear Muon across model sizes.
>
> ---
>
> **Table 2 – LM-Eval accuracies (higher is better)**
>
> | Task | Medium Muon | Medium Muon‑8bit‑lin | Large Muon | Large Muon‑8bit‑lin | XL Muon | XL Muon‑8bit‑lin |
> | --- | --- | --- | --- | --- | --- | --- |
> | ARC‑Challenge | 0.273 | 0.269 | 0.294 | 0.281 | 0.324 | 0.329 |
> | ARC‑Easy | 0.473 | 0.465 | 0.527 | 0.514 | 0.603 | 0.591 |
> | BoolQ | 0.616 | 0.573 | 0.602 | 0.591 | 0.586 | 0.600 |
> | HellaSwag | 0.343 | 0.344 | 0.399 | 0.399 | 0.502 | 0.500 |
> | LAMBADA | 0.236 | 0.227 | 0.312 | 0.299 | 0.387 | 0.384 |
> | MMLU | 0.230 | 0.230 | 0.237 | 0.233 | 0.247 | 0.249 |
> | **Avg. (6 tasks)** | **0.362** | **0.351** | **0.395** | **0.386** | **0.441** | **0.442** |

---

### Meta-Review · Area_Chair_72DJ · 2025-12-24

**Summary:**

This paper proposes a 8-bit muon.

Reviewers concern include novelty, soundness of the theoretical study, and insufficient experimental validation.

**Reviewer Concerns:**

1. novelty: the work is more like a straightforward application of block quantization to muon
2. experiment scale: training token (4B) is too small
3. theoretical soundness and empirical justification of theoretical results.

While the authors argued about the novelty and trained for more tokens (20 token / parameter). Still, the novelty and contribution is rather limited, and the amount of token is still far below modern state-of-the-art. For example, Llama3-8B is trained on 15T tokens, with a token-per-parameter ratio of about 2000.

**Reviewer Scores:**

Original rating is (6,2,2,4), and I think reviewers will retain their original rating.

---

### Decision · Program_Chairs · 2026-01-26

Reject